

# Further development of the reflective practice questionnaire

Shane L. Rogers[1], Lon Van Winkle[2], Nicole Michels[2], Cherie Lucas[3], Hassan Ziada[4], Eduardo Jorge Da Silva[5], Amit Jotangia[6], Sebastian Gabrielsson[7], Silje Gustafsson[7] and Lynn Priddis[8]

[1] Psychology, Edith Cowan University, Perth, Western Australia, Australia
[2] Medical Humanities, Rocky Vista University, Denver, Colorado, United States of America
[3] Pharmacy, University of Technology Sydney, Sydney, New South Wales, Australia
[4] Dental Medicine, University of Nevada, Las Vegas, Nevada, United States of America
[5] Physical Education and Sport, University of Lusofona, Lisbon, Portugal
[6] Cygnet Health Care, Stevenage, United Kingdom
[7] Health, Education and Technology, Lulea University of Technology, Lulea, Sweden
[8] Law School, The University of Western Australia, Perth, Western Australia, Australia

Corresponding author
Shane L. Rogers,
shane.rogers@ecu.edu.au

## ABSTRACT

**Background:** This article provides an update of the Reflective Practice Questionnaire (RPQ). The original RPQ consisted of 40-items with 10-sub-scales. In this article, the RPQ is streamlined into a 10-item single reflective practice construct, and a 30-item extended version that includes additional sub-scales of confidence, uncertainty/ stress, and work satisfaction.

**Methods:** A total of 501 university students filled out an online questionnaire that contained the original Reflective Practice Questionnaire, and two general measures of reflection: The Self-Reflection and Insight Scale, and the Rumination-Reflection Questionnaire.

**Results:** Based on factor analysis, the RPQ was streamlined into a brief 10-item version, and an extended 30-item version. Small positive correlations were found between the RPQ reflective practice measure and the two measures of general reflection, providing discriminant validity evidence for the RPQ. The RPQ was found to be sensitive to differences among industries, whereas the general measures of reflection were not. Average reflective practice scores were higher for health and education industries compared to retail and food/accommodation industries.

## INTRODUCTION

The reflective practice questionnaire (RPQ) was first introduced to the research community as a 40-item questionnaire that contains several sub-scales for assessing self-reported reflective practice and confidence, stress, and work satisfaction (*Priddis & Rogers, 2018*). Following publication, it became apparent from emails of inquiry that many people interested in the measure were practitioners seeking to make use of the RPQ as part of reflective practice initiatives within the workplace. With 40-items across 10 subscales,
the original RPQ provides a broad range of information that can be useful for research studies, however in applied settings people have time and resource constraints that can make such a lengthy questionnaire unwieldy.

Therefore, the primary aim of the current study is to conduct further refinement of the RPQ to reconceptualise the questionnaire as a brief 10-item measure of reflective practice, while also maintaining a longer version of the questionnaire which we re-label as the Reflective Practice Questionnaire—Extended version (RPQ-E). A secondary aim of the study is to examine associations between the RPQ and other general reflection measures to provide evidence that the RPQ provides measurement of reflective practice rather than more generalised reflective tendencies.

## Measuring self-reported reflective practice

The notion of reflective practice is broad, and conceptualisations can vary based on the focus of reflection (*e.g.*, task-focused and/or relational-focused), the context of reflection (*e.g.*, work context *vs* learning context), when it occurs (*e.g.*, during action *vs* after action), with who it occurs (*e.g.*, self-reflection *vs* reflection with others), and how it occurs (*e.g.*, meditative *vs* critical reflection) (*Greenberger, 2020*; *Hebert, 2015*; *Mezirow, 1991*; *Ooi, Fisher & Coker, 2021*; *Schon, 1995*; *Thompson & Pascal, 2012*; *Tsingos, Bosnic-Anticevich & Smith, 2014*). In this article our conceptualisation of reflective practice as measured by the reflective practice questionnaire can be described as the tendency to actively reflect upon the thoughts and actions that occur when working with clients. These reflections might be about relational aspects of working with clients (*e.g.*, Are they or I frustrated?), or more task focused (*e.g.*, Are we making good progress?). Reflections can potentially occur in-the-moment during interaction (*i.e.*, reflection-in-action) or sometime after the interaction has occurred (*i.e.*, reflection-on-action). Reflections can be about one's own thoughts/actions and/or those of the client/s. The reflections can be either more meditative in nature (*i.e.*, wondering with simple curiosity) or more critical (*i.e.*, critically questioning ways of thinking/doing).

## The reflective practice questionnaire (RPQ)

The RPQ was originally designed as an instrument to measure self-reported reflective practice alongside several other variables that have relevance for reflective practice: desire for improvement, general confidence, communication confidence, uncertainty, stress, and work satisfaction (*Priddis & Rogers, 2018*). The RPQ sets itself apart from other self-report reflection measures by predominately focusing on working with clients, and by utilising broad phrasing so that the measure can be used across a wide range of professions where reflective practice is relevant (For a discussion, see: *Priddis & Rogers, 2018*).

Studies have been conducted utilising the RPQ with medical students (*Bari et al., 2021*; *Horst et al., 2019*; *Khoshgoftar & Barkhordari-Sharifabad, 2023a, 2023b*; *Lee et al., 2023*; *Rogers et al., 2019*; *Schwartz et al., 2020*; *Van Winkle et al., 2021, 2022*), surgeons/physicians (*Aitken et al., 2021*; *Whelehan, Conlon & Ridgway, 2021*), nurses (*Aitken et al., 2021*; *Al-Osaimi, 2022*; *Gabrielsson et al., 2022*; *Gustafsson et al., 2020*; *Khalil & Hashish, 2022*), psychologists (*Sadusky & Spinks, 2022*), allied health

professionals (*Aurora, Mawren & Fullam, 2023*; *Or & Golba, 2023*; *Parrott et al., 2023*), pre-service teachers (*Day, Webster & Killen, 2022*; *Fuertes-Camacho, Dulsat-Ortiz & Alvarez-Canocas, 2021*), qualified teachers (*Chen & Chen, 2022*; *Gross, 2020*; *Moeder-Chandler, 2020*), and sport coaches (*Da Silva et al., 2022*). In these studies the RPQ has been used for a range of purposes, such as assessment of the reliability of the RPQ scales (*e.g.*, *Gustafsson et al., 2020*), comparison between different sub-groups of participants (*e.g.*, *Day, Webster & Killen, 2022*), and comparison across different time points to explore student development (*e.g.*, *Van Winkle et al., 2021*).

*Van Winkle et al. (2021)* have published work that demonstrates how the RPQ can be used as part of an evaluation of teaching methods (*Horst et al., 2019*; *Schwartz et al., 2020*; *Van Winkle et al., 2021*, *2022*). For example, *Van Winkle et al. (2021)* found that self-reported reflective practice and a self-report empathy measure significantly increased for most medical students enrolled in a 4-month online course that included activities designed to facilitate the development of reflective practice. In another example, *Van Winkle et al. (2022)* found that the magnitude of increase in self-reported reflective practice and empathy was higher for prospective medical students who completed a course that included reflection specifically on their service-learning activities compared to students that completed a similar course with reflection only upon conflict interactions outside of their work environment.

Several other scholars have also made use of the RPQ when evaluating learning activities (*Da Silva et al., 2022*; *Khalil & Hashish, 2022*). *Da Silva et al. (2022)* found that self-reported reflective practice was higher for a group of sport coaches that underwent a reflective journalling intervention compared with a control group. *Khalil & Hashish (2022)* found that average self-reported reflective practice increased after reflective practice training, and that self-reported reflective practice was positively associated with self-reported critical thinking tendencies.

## The present study—considerations for further development of the reflective practice questionnaire

Since the initial publication of the RPQ in 2018, correspondence received from researchers and practitioners has informed our reflections on how the RPQ might best serve the community that uses it. In our initial development of the RPQ we were interested in developing a comprehensive questionnaire. The RPQ was published with ten sub-scales, five that were focused on elements of reflective practice (*i.e.*, reflection in action, reflection on action, self-appraisal, and reflection with others) with the remaining six sub-scales focused on other constructs of relevance to reflective practice (*i.e.*, desire for improvement, general confidence, confidence in communication, uncertainty, stress interacting with clients, and job satisfaction).

Something that became apparent to us was that perhaps the RPQ contained too many sub-scales. Both researchers and practitioners were most interested in a simple and clear measure of self-reported reflective practice. In response to this we published a follow up article in 2019 proposing a single reflective practice score by averaging across the four reflective practice sub-scales of the RPQ (*Rogers et al., 2019*). We were not surprised to see

most of the subsequent studies utilising the RPQ made use of this more simplified conceptualisation of the reflective practice measure (*Al-Osaimi, 2022*; *Bari et al., 2021*; *Da Silva et al., 2022*; *Day, Webster & Killen, 2022*; *Gabrielsson et al., 2022*; *Gross, 2020*; *Gustafsson et al., 2020*; *Horst et al., 2019*; *Khalil & Hashish, 2022*; *Or & Golba, 2023*; *Schwartz et al., 2020*; *Van Winkle et al., 2021*, *2022*; *Whelehan, Conlon & Ridgway, 2021*).

Considering that the use of an overall single reflective practice score has emerged as the most popular usage for the RPQ, we felt that it would be worthwhile investigating the scope for a shorter version of the combined RPQ reflective sub-scales. We also felt it was desirable to revisit the other sub-scales within the original RPQ to explore if some aggregation across the sub-scales might be statistically justifiable. Therefore, a primary aim of the present study was to explore if the RPQ structure could be simplified. We utilised factor analytic techniques to achieve this aim.

A secondary aim of the present study was to examine if the measure of self-reported reflective practice obtained by the RPQ is best conceptualised as a distinct and separate construct from a broader notion of self-reflection. The RPQ was designed as a measure specifically targeted on the act of reflection in work practice with clients. However, the RPQ has not previously been compared to more general trait-based measures of self-reflection. In the present study we compare the RPQ with two well-cited general measures of self-reflection, the Self-Reflection and Insight Scale (*Grant, Franklin & Langford, 2002*), and the Rumination-Reflection Questionnaire (*Trapnell & Campbell, 1999*).

## MATERIALS AND METHODS

### Participants

Prior to conducting this research ethical approval was obtained from the Edith Cowan University ethics review board. Ethics reference number: 2019-00741-ROGERS. Five hundred and one undergraduate psychology students participated in this study for 0.5 credit points for a research participation component in a statistics unit. A requirement for participation was that the person must be currently employed in paid work in addition to their university studies. Research consent was obtained in a check box as part of the online survey. The main industries that participants indicated they worked in were in retail (25%), health care and social assistance (19%), education and training (13%), and accommodation and food services (13%). The remaining 30% worked in other miscellaneous industries. All participants indicated that they interact with clients at least once a month, with a specific breakdown: Every day (81%), every few days (14%), about once a week (3%), about once a fortnight (1%), and about once a month (1%).

### Measures

Each participant answered the 40-item *Reflective Practice Questionnaire* (*Priddis & Rogers, 2018*). In this study we changed the response scale from the original six-point Not at all—Extremely scale to be a six-point Very rarely—Almost always scale (scoring: 1. Very rarely 2. Rarely 3. Sometimes 4. Often 5. Very often 6. Almost always). Some minor modifications were made to the individual items of the questionnaire to account for the change in response scale. After sub-scales were determined *via* the factor analysis, sub-scale scores

were calculated *via* averaging across relevant items. A brief evaluation study examining the change of response scale from the original RPQ can be found as a document titled 'RPQ response scale evaluation' alongside the raw data for this article at: https://doi.org/10.6084/m9.figshare.22776251.v1.

Two other questionnaires were used in this study: The 20-item Self-Reflection and Insight Scale (SRIS) (*Grant, Franklin & Langford, 2002*), and the 24-item Rumination-Reflection Questionnaire (RRQ) (*Trapnell & Campbell, 1999*). Prior studies have consistently reported good reliability values for both questionnaires (*DaSilveira, DeSouza & Gomes, 2015*; *Grant, Franklin & Langford, 2002*; *Harrington & Loffredo, 2010*; *Trapnell & Campbell, 1999*).

The SRIS contains two sub-scales, the Self-reflection sub-scale (note, this sub-scale is comprised of two strongly correlated sub-facets: Engagement in self-reflection, for example: "I frequently take time to reflect on my thoughts", and need for self-reflection, for example "It is important for me to evaluate the things that I do"), and the Insight sub-scale, for example "I usually have a very clear idea about why I've behaved in a certain way" (*Grant, Franklin & Langford, 2002*). When answering the SRIS participants were asked "Please rate your level of disagreement/agreement for each statement on a scale that ranges from (1) Strongly disagree to (6) Strongly agree". In between the two poles (*i.e.*, 1 and 6) the numbers (2), (3), (4), and (5) were presented as options. Sub-scale scores were calculated by averaging across relevant items.

The RRQ contains two sub-scales, Rumination, for example "I often reflect on episodes of my life that I should no longer concern myself with", and Reflection, for example "I love analysing why I do things" (*Trapnell & Campbell, 1999*). When answering the RRQ participants were asked "Please rate your level of disagreement/agreement for each statement". The response scale used was (1) Strongly disagree, (2) Disagree, (3) Neutral, (4) Agree, (5) Strongly agree. Sub-scale scores were calculated by averaging across relevant items.

## RESULTS

### Factor analysis of the updated RPQ—the reflective practice scale

The raw data for this manuscript is available at: https://doi.org/10.6084/m9.figshare.22776251.v1. An exploratory factor analysis was conducted on the 16-items of the RPQ that prior studies have previously combined to provide a 'reflective capacity' measure. These items consisted of the reflection-in-action, reflection-on-action, reflection-with-others and self-appraisal sub-scales from the original RPQ. The exploratory factor analysis was conducted using the statistical software Stata, using the principal factors method, applying an oblique Promax rotation. Two factors had an eigenvalue greater than 1 (*i.e.*, factor 1 = 6.19, factor 2 = 1.32). The factor loadings from this analysis are presented in Table 1.

The reflection-with-others (RO) items loaded onto the second factor. There are two reasons we suggest this might be the case. First, a point of difference between the RO items and all others is that the wording of the RO items lacks specific reference to working with clients, and instead refers simply to 'work'. This may lead some participants to interpret

**Table 1 Factor loadings from exploratory factor analysis on the reflective practice items from the original reflective practice questionnaire.** Loadings less than 0.40 are omitted for clarity.

| Item | Factor 1. | Factor 2. | Uniqueness. |
|---|---|---|---|
| 1. (RiA) During interactions with clients I recognize when my pre-existing beliefs are influencing the interaction. | 0.59 | | 0.69 |
| 2. (RiA) During interactions with clients I consider how my personal thoughts and feelings are influencing the interaction. | 0.77 | | 0.46 |
| 3. (RiA) During interactions with clients I recognize when their pre-existing beliefs are influencing the interaction. | 0.60 | | 0.60 |
| 4. (RiA) During interactions with clients I consider how their personal thoughts and feelings are influencing the interaction. | 0.68 | | 0.54 |
| 5. (RoA) After interacting with clients I spend time thinking about what was said and done. | 0.65 | | 0.57 |
| 6. (RoA) After interacting with clients I wonder about the client's experience of the interaction. | 0.75 | | 0.52 |
| 7. (RoA) After interacting with clients I wonder about my own experience of the interaction. | 0.71 | | 0.47 |
| 8. (RoA) After interacting with clients I think about how things went during the interaction. | 0.80 | | 0.37 |
| 9. (RO) When reflecting with others about my work I become aware of things I had not previously considered. | | 0.63 | 0.61 |
| 10. (RO) When reflecting with others about my work I develop new perspectives. | | 0.79 | 0.42 |
| 11. (RO) Reflecting with others about my work helps me to work out problems. | | 0.68 | 0.50 |
| 12. (RO) I gain new insights when reflecting with others about my work. | | 0.83 | 0.35 |
| 13. (SA) I think about my strengths for working with clients. | 0.31 | | 0.74 |
| 14. (SA) I think about my weaknesses for working with clients. | 0.56 | | 0.61 |
| 15. (SA) I think about how I might improve my ability to work with clients. | 0.44 | | 0.52 |
| 16. (SA) I critically evaluate the strategies and techniques I use in my work with clients. | 0.58 | | 0.51 |

**Note:**
Sub-scale items from the original RPQ: RiA, reflection-in-action; RoA, reflection-on-action; RO, reflection-with-others; SA, self-appraisal.

these items in a broader sense in comparison to other items. Second, the RO items are specific to the notion of reflecting with others, whereas all other items make no explicit mention of others. Based on the factor analysis result, we made the decision to cut-down the RPQ reflection measure by removal of the RO items.

The removal of the RO items reduces the item count from 16 to 12. We noticed an item from the self-appraisal scale had a lower than ideal factor loading of 0.31 (*i.e.*, "I think about my strengths for working with clients"). Therefore, we decided that removal of that item was justifiable, and we also decided on removal of the other self-appraisal item about weaknesses (*i.e.*, "I think about my weaknesses for working with clients") so that the remaining two self-appraisal items contain consistent general phrasing (*i.e.*, "I think about how I might improve my ability to work with clients" and "I critically evaluate the strategies and techniques I use in my work with clients"). This results in 10 items for our proposed 'reflective practice' scale to represent the core scale of the reenvisaged RPQ.

## Factor analysis of the updated RPQ—the extended version of the RPQ

A follow up exploratory factor analysis was conducted to further examine consolidation of the extended form of the RPQ. In this analysis we included the 10 reflective practice items from the prior analysis alongside all other items from the original RPQ. An exception was the desire for improvement items that we left out of the analysis. This was left out because after careful consideration we determined that the original RPQ sub-scale does not provide

**Table 2 Exploratory factor analysis of the reflective practice questionnaire—extended (RPQ-E).** Loadings less than 0.40 are omitted for clarity.

| Item | Factor loadings | | | | Uniqueness |
|---|---|---|---|---|---|
| | 1. | 2. | 3. | 4. | |
| 1. (RiA) During interactions with clients I recognize when my pre-existing beliefs are influencing the interaction. | 0.48 | | | | 0.71 |
| 2. (RiA) During interactions with clients I consider how my personal thoughts and feelings are influencing the interaction. | 0.69 | | | | 0.51 |
| 3. (RiA) During interactions with clients I recognize when their pre-existing beliefs are influencing the interaction. | 0.63 | | | | 0.59 |
| 4. (RiA) During interactions with clients I consider how their personal thoughts and feelings are influencing the interaction. | 0.77 | | | | 0.48 |
| 5. (RoA) After interacting with clients I spend time thinking about what was said and done. | 0.63 | | | | 0.58 |
| 6. (RoA) After interacting with clients I wonder about the client's experience of the interaction. | 0.69 | | | | 0.52 |
| 7. (RoA) After interacting with clients I wonder about my own experience of the interaction. | 0.72 | | | | 0.48 |
| 8. (RoA) After interacting with clients I think about how things went during the interaction. | 0.80 | | | | 0.38 |
| 9. (SA) I think about how I might improve my ability to work with clients. | 0.53 | | | | 0.52 |
| 10. (SA) I critically evaluate the strategies and techniques I use in my work with clients. | 0.67 | | | | 0.44 |
| 11. (CG) I feel like I have all the experience I require to effectively interact with clients. | | 0.78 | | | 0.46 |
| 12. (CG) I feel like I have all the practical skills I require to effectively interact with clients. | | 0.86 | | | 0.32 |
| 13. (CG) I feel like I have learnt everything I need to know in order to effectively interact with clients. | | 0.66 | | | 0.61 |
| 14. (CG) I feel like I have all the theoretical knowledge I require to effectively interact with clients. | | 0.72 | | | 0.53 |
| 15. (CC) I feel able to communicate so that a client can understand me easily. | | 0.60 | | | 0.52 |
| 16. (CC) I feel confident when communicating my ideas with a client. | | 0.46 | | | 0.49 |
| 17. (CC) I feel that I provide clear messages to my clients. | | 0.58 | | | 0.48 |
| 18. (CC) I feel capable in my ability to communicate with clients. | | 0.63 | | | 0.35 |
| 19. (UNC) I am uncertain that my planning for a client is the best possible way to proceed. | | | 0.47 | | 0.75 |
| 20. (UNC) I am uncertain that I am interpreting the needs of a client correctly. | | | 0.50 | | 0.61 |
| 21. (UNC) I am uncertain about how to handle the needs of a client. | | | 0.61 | | 0.48 |
| 22. (UNC) I am uncertain that I properly understand the needs of a client. | | | 0.51 | | 0.64 |
| 19. (STR) After interacting with clients I feel exhausted. | | | 0.65 | | 0.59 |
| 20. (STR) I find interacting with a client to be stressful. | | | 0.75 | | 0.42 |
| 21. (STR) I feel distressed after communicating with a client. | | | 0.70 | | 0.52 |
| 22. (STR) The pressure to meet needs of a client can feel overwhelming. | | | 0.62 | | 0.58 |
| 27. (JS) My work provides me with a sense of fulfilment. | | | | 0.87 | 0.23 |
| 28. (JS) I feel like my work means more to me than simply earning money. | | | | 0.79 | 0.33 |
| 29. (JS) I enjoy my work. | | | | 0.89 | 0.22 |
| 30. (JS) I find my work rewarding. | | | | 0.59 | 0.57 |

Note:
Sub-scale items from the original RPQ: RiA, reflection-in-action; RoA, reflection-on-action; RO, reflection-with-others; SA, self-appraisal; CG, confidence-general; CC, confidence-communication; UNC, uncertainty; STR, stress; JS, job satisfaction.

a measure that is nuanced enough to adequately cover this complex motivational construct (*Breckenridge et al., 2019*; *Leach & Iyer, 2023*).

The same type of exploratory factor analysis was conducted as the prior analysis, using the principal factors method, applying an oblique Promax rotation. Four factors had an eigenvalue greater than 1 (*i.e.*, factor 1 = 5.93, factor 2 = 5.48. factor 3 = 2.53, factor

**Table 3** Means (with standard deviation in brackets) for the sub-scales of the RPQ-E, SRIS, and RRQ separated by industry groups.

| | Health | Education | Admin. | Retail | Acomm./Food | Other | Total | Cronbach's alpha |
|---|---|---|---|---|---|---|---|---|
| RPQ. Reflective practice | 4.03 (0.78) | 3.98 (0.74) | 3.67 (0.90) | 3.65 (0.86) | 3.46 (0.89) | 3.77 (0.82) | 3.76 (0.85) | 0.89 |
| RPQ. Confidence | 4.08 (0.75) | 4.19 (0.74) | 4.25 (0.93) | 4.37 (0.84) | 4.01 (0.76) | 4.26 (0.78) | 4.21 (0.81) | 0.87 |
| RPQ. Uncertainty/Stress | 2.67 (0.77) | 2.77 (0.83) | 2.52 (0.73) | 2.75 (0.75) | 2.71 (0.75) | 2.66 (0.78) | 2.69 (0.77) | 0.83 |
| RPQ. Work satisfaction | 4.61 (1.00) | 4.72 (0.94) | 4.13 (1.11) | 3.61 (1.14) | 3.53 (0.93) | 3.92 (1.18) | 4.05 (1.15) | 0.86 |
| SRIS. Self-reflection | 4.85 (0.82) | 4.73 (0.96) | 4.92 (0.88) | 4.68 (0.99) | 4.68 (1.03) | 4.92 (0.82) | 4.79 (0.92) | 0.93 |
| SRIS. Insight | 4.28 (0.86) | 4.30 (0.94) | 4.64 (0.87) | 4.02 (0.92) | 4.01 (0.89) | 4.44 (0.89) | 4.25 (0.92) | 0.85 |
| RRQ. Reflection | 3.58 (0.68) | 3.53 (0.79) | 3.61 (0.90) | 3.59 (0.76) | 3.57 (0.67) | 3.82 (0.79) | 3.62 (0.77) | 0.93 |
| RRQ. Rumination | 3.59 (0.68) | 3.65 (0.90) | 3.69 (0.81) | 3.69 (0.73) | 3.70 (0.74) | 3.57 (0.83) | 3.65 (0.77) | 0.93 |

4 = 1.15). The rotated factor loadings from this analysis are presented in Table 2. These factors represent reflective practice, confidence, uncertainty/stress, and work satisfaction.

## Comparisons among industry means

For all measures we compared across the different industry groups by running a series of one-way ANOVAs with Follow up Bonferroni adjusted comparisons. We excluded the 'other' category when running the analyses. An overall difference among reflective practice means was found, $F(4,405) = 6.60$, $p < 0.001$, $\eta_p^2 = 0.06$, see Table 3. Follow up comparisons revealed that this result was due to the Health and Education profession means being significantly higher than the retail and accommodation & food means ($ps < 0.05$, although note education & retail comparison $p = 0.08$). There was no difference among the administration, retail, and accommodation & food groups ($ps > 0.05$). Nor was there any difference between the health and education groups ($p > 0.05$).

There was an overall difference found among RPQ confidence means, $F(4,405) = 2.88$, $p = 0.02$, $\eta_p^2 = 0.03$, however this was due to a marginally significant difference only between the retail and accommodation/food mean ($p = 0.04$). There was no significant difference among RPQ uncertainty/stress means, $F(4,405) = 1.20$, $p = 0.31$. There was an overall difference found among RPQ work satisfaction means, $F(4,405) = 23.13$, $p < 0.001$, $\eta_p^2 = 0.19$, with follow up comparisons revealing statistical differences among means followed the pattern: Health = Education > Administration > Retail = Accommodation/food.

For SRIS self-reflection there was no difference among the industry means, $F(4,405) = 0.99$, $p = 0.41$. There was an overall difference among means for SRIS insight, $F(4,405) = 6.01$, $p < 0.001$, $\eta_p^2 = 0.06$, with follow up comparisons revealing that this result was due to the Administration industry mean higher than the retail and accommodation & food means ($ps < 0.001$), with all other comparisons non-significant. There was no difference among profession means for RRQ self-reflection ($F(4,405) = 0.10$, $p = 0.98$), or rumination, $F(4,405) = 0.32$, $p = 0.87$.

Rogers et al. (2024), *PeerJ*, DOI 10.7717/peerj.16879

**Table 4 Pearson correlations among the sub-scales of the RPQ-E, SRIS, and RRQ.**

|  | RPQ. Ref. Prac. | RPQ. Conf. | RPQ. Unc./Stress | RPQ. Work Satisfaction | SRIS. Self-reflection | SRIS. Insight | RRQ. Reflection | RRQ. Rumination |
|---|---|---|---|---|---|---|---|---|
| RPQ. Ref. Prac. | 1 | | | | | | | |
| RPQ. Conf. | 0.17* | 1 | | | | | | |
| RPQ. Unc./Stress | 0.26* | −0.40* | 1 | | | | | |
| RPQ. Work satisfaction | 0.30* | 0.21* | −0.22* | 1 | | | | |
| SRIS. Self-reflection | 0.32* | 0.13* | −0.06 | 0.09* | 1 | | | |
| SRIS. Insight | 0.06 | 0.27* | −0.40* | 0.20* | 0.32* | 1 | | |
| RRQ. Reflection | 0.23* | 0.03 | −0.03 | 0.10* | 0.71* | 0.26* | 1 | |
| RRQ. Rumination | 0.14* | −0.19* | 0.38* | −0.22* | 0.19* | −0.36* | 0.11* | 1 |

Note:
* $p < 0.05$.

## Correlations between the RPQ, SRIS, and RRQ

Correlations among all measures are presented in Table 4. Of particular interest are the correlations between the RPQ reflective practice measure with the SRIS self-reflection ($r = 0.32$, $p < 0.05$) and RRQ self-reflection ($r = 0.23$, $p < 0.05$) measures. Both associations are of relatively weak magnitude. To double check that these associations are not the result of analysing a sample where people from different industries are lumped together, we checked the correlations after splitting the datafile by industry group. This did not change the overall result, with the correlation between RPQ reflective practice and SRIS self-reflection ranging from 0.12–0.48, and the correlation between RPQ reflective practice and RRQ self-reflection ranging from 0.09–0.45, across the industry groups.

## DISCUSSION

In this study we propose a revision of the Reflective Practice Questionnaire (RPQ) that was originally published by *Priddis & Rogers (2018)*. Guided by factor analysis results, we propose a revised 10-item version of the RPQ that provides a self-report measure of reflective practice, see Table 5. We also propose a 30-item version of the questionnaire that we call the RPQ extended (RPQ-E), see Table 6. This version contains the 10-item reflective practice scale along with additional sub-scales for confidence, uncertainty/stress, and work satisfaction. A secondary aim was to compare the RPQ with two general measures of self-reflection to test if the RPQ can be considered as providing a measure of reflective practice that is distinct from general reflection measures. We found low correlations between the RPQ and the general self-reflection measures that provides support for this assertion.

### Modification of the RPQ

An initial overall change from the original RPQ is to change the response scale from a six-point 'Not at all—Extremely' to a six-point 'Very rarely—Almost always' Likert-type scale. The reasoning behind this decision is that asking participants the extent that they

**Table 5 The reflective practice questionnaire (RPQ).**

| | (1). Very rarely | (2). Rarely | (3). Sometimes | (4). Often | (5). Very often | (6). Almost always |
|---|---|---|---|---|---|---|
| 1. During interactions with clients I recognize when my pre-existing beliefs are influencing the interaction. | ☐ | ☐ | ☐ | ☐ | ☐ | ☐ |
| 2. During interactions with clients I consider how my personal thoughts and feelings are influencing the interaction. | ☐ | ☐ | ☐ | ☐ | ☐ | ☐ |
| 3. During interactions with clients I recognize when their pre-existing beliefs are influencing the interaction. | ☐ | ☐ | ☐ | ☐ | ☐ | ☐ |
| 4. During interactions with clients I consider how their personal thoughts and feelings are influencing the interaction. | ☐ | ☐ | ☐ | ☐ | ☐ | ☐ |
| 5. After interacting with clients I spend time thinking about what was said and done | ☐ | ☐ | ☐ | ☐ | ☐ | ☐ |
| 6. After interacting with clients I wonder about the client's experience of the interaction. | ☐ | ☐ | ☐ | ☐ | ☐ | ☐ |
| 7. After interacting with clients I wonder about my own experience of the interaction. | ☐ | ☐ | ☐ | ☐ | ☐ | ☐ |
| 8. After interacting with clients I think about how things went during the interaction. | ☐ | ☐ | ☐ | ☐ | ☐ | ☐ |
| 9. I think about how I might improve my ability to work with clients. | ☐ | ☐ | ☐ | ☐ | ☐ | ☐ |
| 10. I critically evaluate the strategies and techniques I use in my work with clients. | ☐ | ☐ | ☐ | ☐ | ☐ | ☐ |

**Note:**
Scoring instructions, average across all items to obtain a score that can potentially range from 1–6.

**Table 6 The reflective practice questionnaire extended (RPQ-E).**

| | (1). Very rarely | (2). Rarely | (3). Sometimes | (4). Often | (5). Very often | (6). Almost always |
|---|---|---|---|---|---|---|
| 1. During interactions with clients I recognize when my pre-existing beliefs are influencing the interaction. | ☐ | ☐ | ☐ | ☐ | ☐ | ☐ |
| 2. During interactions with clients I consider how my personal thoughts and feelings are influencing the interaction. | ☐ | ☐ | ☐ | ☐ | ☐ | ☐ |
| 3. During interactions with clients I recognize when their pre-existing beliefs are influencing the interaction. | ☐ | ☐ | ☐ | ☐ | ☐ | ☐ |
| 4. During interactions with clients I consider how their personal thoughts and feelings are influencing the interaction. | ☐ | ☐ | ☐ | ☐ | ☐ | ☐ |
| 5. After interacting with clients I spend time thinking about what was said and done | ☐ | ☐ | ☐ | ☐ | ☐ | ☐ |
| 6. After interacting with clients I wonder about the client's experience of the interaction. | ☐ | ☐ | ☐ | ☐ | ☐ | ☐ |
| 7. After interacting with clients I wonder about my own experience of the interaction. | ☐ | ☐ | ☐ | ☐ | ☐ | ☐ |
| 8. After interacting with clients I think about how things went during the interaction. | ☐ | ☐ | ☐ | ☐ | ☐ | ☐ |
| 9. I think about how I might improve my ability to work with clients. | ☐ | ☐ | ☐ | ☐ | ☐ | ☐ |
| 10. I critically evaluate the strategies and techniques I use in my work with clients. | ☐ | ☐ | ☐ | ☐ | ☐ | ☐ |

| | (1). Very rarely | (2). Rarely | (3). Sometimes | (4). Often | (5). Very often | (6). Almost always |
|---|---|---|---|---|---|---|
| 11. I feel like I have all the experience I require to effectively interact with clients. | ☐ | ☐ | ☐ | ☐ | ☐ | ☐ |
| 12. I feel like I have all the practical skills I require to effectively interact with clients. | ☐ | ☐ | ☐ | ☐ | ☐ | ☐ |
| 13. I feel like I have learnt everything I need to know in order to effectively interact with clients. | ☐ | ☐ | ☐ | ☐ | ☐ | ☐ |
| 14. I feel like I have all the theoretical knowledge I require to effectively interact with clients. | ☐ | ☐ | ☐ | ☐ | ☐ | ☐ |
| 15. I feel able to communicate so that a client can understand me easily. | ☐ | ☐ | ☐ | ☐ | ☐ | ☐ |
| 16. I feel confident when communicating my ideas with a client. | ☐ | ☐ | ☐ | ☐ | ☐ | ☐ |
| 17. I feel that I provide clear messages to my clients. | ☐ | ☐ | ☐ | ☐ | ☐ | ☐ |
| 18. I feel capable in my ability to communicate with clients. | ☐ | ☐ | ☐ | ☐ | ☐ | ☐ |
| 19. After interacting with clients I feel exhausted. | ☐ | ☐ | ☐ | ☐ | ☐ | ☐ |
| 20. I find interacting with a client to be stressful. | ☐ | ☐ | ☐ | ☐ | ☐ | ☐ |
| 21. I feel distressed after communicating with a client. | ☐ | ☐ | ☐ | ☐ | ☐ | ☐ |
| 22. The pressure to meet needs of a client can feel overwhelming. | ☐ | ☐ | ☐ | ☐ | ☐ | ☐ |
| 23. I am uncertain that my planning for a client is the best possible way to proceed. | ☐ | ☐ | ☐ | ☐ | ☐ | ☐ |
| 24. I am uncertain that I am interpreting the needs of a client correctly. | ☐ | ☐ | ☐ | ☐ | ☐ | ☐ |
| 25. I am uncertain about how to handle the needs of a client. | ☐ | ☐ | ☐ | ☐ | ☐ | ☐ |
| 26. I am uncertain that I properly understand the needs of a client. | ☐ | ☐ | ☐ | ☐ | ☐ | ☐ |
| 27. My work provides me with a sense of fulfilment. | ☐ | ☐ | ☐ | ☐ | ☐ | ☐ |
| 28. I feel like my work means more to me than simply earning money. | ☐ | ☐ | ☐ | ☐ | ☐ | ☐ |
| 29. I enjoy my work. | ☐ | ☐ | ☐ | ☐ | ☐ | ☐ |
| 30. I find my work rewarding. | ☐ | ☐ | ☐ | ☐ | ☐ | ☐ |

**Notes:**

Scoring, all measures provide a score that can range from 1–6. Reflective practice score = Average across items 1–10. Confidence score = Average across items 11–18. Uncertainty/Stress score = Average across items 19–26. Work satisfaction score = Average across items 27–30.

There is scope for the confidence sub-scale to be further broken down into 'general confidence' (items 11–14) and 'communication confidence' (items 15–18) sub-scales. There is scope for the uncertainty/stress sub-scale to be further broken down into 'stress' (items 19–22) and 'uncertainty' (items 23–26) sub-scales.

engage in reflective practice might be confusing for some participants. For example, a participant might not fully understand the difference between being reflective 'moderately' *vs* 'very much'. It should be easier for a participant to decide upon how often they reflect on their thought and behaviours. We concede there might still be some uncertainty, for example deciding between 'sometimes' *vs* 'often', however we believe this still constitutes an improvement over the original response scale.

Most research studies to date using the RPQ have averaged across the original RPQ sub-scales 'reflection-in-action', 'reflection-on-action', 'reflection with others', and 'self-appraisal' for a 16-item measure of reflective practice (*Al-Osaimi, 2022*; *Bari et al., 2021*; *Da Silva et al., 2022*; *Day, Webster & Killen, 2022*; *Gabrielsson et al., 2022*; *Gross, 2020*; *Gustafsson et al., 2020*; *Horst et al., 2019*; *Khalil & Hashish, 2022*; *Schwartz et al., 2020*; *Van Winkle et al., 2021*, *2022*; *Whelehan, Conlon & Ridgway, 2021*). In the present study, an exploratory factor analysis revealed that the 'reflection with others' items loaded onto a

separate factor, so these were dropped. We also made the decision to drop an item from the 'self-appraisal' items with a low loading on the reflective practice primary factor. We also dropped one more of the 'self-appraisal' items to bring the measure down to 10 items to make it easier for averaging items to create the overall score. We expect these changes will make using the RPQ more user friendly, especially in applied settings.

We also used factor analysis results to inform decision making to simplify the sub-scales of the extended version of the RPQ to include 'confidence', 'uncertainty/stress', and 'work satisfaction', alongside the 10-item 'reflective practice' component. The full extended version of the RPQ has therefore changed from the original 40-item questionnaire with 10 sub-scales to a 30-item questionnaire with four sub-scales. We expect these changes will make the extended version of the RPQ more user friendly.

## Comparing the RPQ with general measures of reflection

Testing for discriminant validity (also called divergent validity) is an important part of questionnaire evaluation (*Fiske, 1982*; *Lucas, Diener & Suh, 1996*). Evidence for discriminant validity is obtained when one measure has only weak association with another measure that is theoretically not expected to overlap with the main measure of interest. For example, in an evaluation study of the Self-Reflection and Insight Scale (SRIS), *Banner et al. (2023)* tested for discriminant validity by comparing the SRIS with the perceived knowledge subscale from the short form of the Career Futures Inventory (*McIlveen, Burton & Beccaria, 2013*). *Banner et al. (2023)* reported no statistically significant correlation between these measures, concluding this represented some discriminant validity evidence for the SRIS.

An additional aim of the present study was to contrast the RPQ, a measure designed to specifically measure self-reflection upon one's work, with more general measures of self-reflection. The goal was to provide some evidence that the RPQ reflective practice measure provides a measure that can be differentiated from more general self-reflective tendencies of an individual (*i.e.*, discriminant validity evidence). We therefore included two well-cited general measures of self-reflection in our study, the Self-Reflection and Insight Scale (SRIS) (*Grant, Franklin & Langford, 2002*), and the Rumination-Reflection Questionnaire (RRQ) (*Trapnell & Campbell, 1999*). As expected, the RPQ reflective practice score was found to only have weak positive associations with these measures, suggesting that it does measure a different construct.

Additionally, the RPQ reflective practice mean was found to be significantly higher for participants in the healthcare and education industries compared with other industries such as retail and food/accommodation. This is consistent with *Priddis & Rogers (2018)* original findings and is consistent with the intuitive notion that reflective practice would be higher in workplaces where reflective practice is encouraged and/or explicitly taught as part of qualifications. The SRIS and RRQ general self-reflection measures did not differ across the industry groups. This provides some further evidence for the validity of the RPQ as a measure of reflective practice.

## Limitations and future research

An inherent limitation associated with the RPQ is the self-report nature of the measure. Just because a person thinks they are very reflective, does not guarantee this to be true. Any self-report measures of reflection should be used with this in mind, and thus used with caution. However, this does not invalidate the use of such measures. As reviewed in our introduction to this article, evidence does exist suggesting that the RPQ can be sensitive to changes in reflective practice tendencies of individuals (*Aitken et al., 2021*; *Da Silva et al., 2022*; *Horst et al., 2019*; *Khalil & Hashish, 2022*; *Schwartz et al., 2020*; *Van Winkle et al., 2021*, *2022*).

Another limitation of the present study is the reliance on a convenience sample of university students. We were originally planning on having several participant groups; however, the COVID-19 pandemic introduced challenges for that data collection. Regardless, the sample we obtained is serviceable for the purposes of the current article. In future research we will continue validation work of the RPQ across different samples, and for different applications of the RPQ. Introducing the more user-friendly version of the RPQ in this current article we expect will help facilitate that process.

While we believe the refinement of the RPQ as presented in this article is a step forward in the development of the questionnaire, we also recognise that simplifying the questionnaire may not be beneficial for all potential applications of the questionnaire. For example, *Sadusky & Spinks (2022)* reported that burnout was associated with the stress sub-scale of the original RPQ, but not with the uncertainty sub-scale. Therefore, research questions that dig deeper into the sub-aspects contained with the RPQ may benefit from using the original version of the RPQ or breaking down the combined sub-scales of the updated RPQ (*e.g.*, separating the uncertainty/stress sub-scale into separate uncertainty and stress scores).

## CONCLUSIONS

The purpose of the current study was to further refine the reflective practice questionnaire with the intention of making it more streamlined. In this article we provide a slightly modified version of the RPQ (see Tables 5 and 6) that we believe will make it a more user-friendly questionnaire for both researchers and practitioners. The RPQ is free to use and there is no requirement to obtain permission from the authors for use. However, we do enjoy hearing from people about how they are using it and are always happy to receive emails letting us know what you are using it for, or any questions you may have.

### Funding

The authors received no funding for this work.

### Competing Interests

Shane L. Rogers is an Academic Editor for PeerJ. Amit Jotangia is employed by Cygnet Health Care.

## Author Contributions

- Shane L. Rogers conceived and designed the experiments, performed the experiments, analyzed the data, prepared figures and/or tables, authored or reviewed drafts of the article, and approved the final draft.
- Lon Van Winkle conceived and designed the experiments, authored or reviewed drafts of the article, and approved the final draft.
- Nicole Michels conceived and designed the experiments, authored or reviewed drafts of the article, and approved the final draft.
- Cherie Lucas conceived and designed the experiments, authored or reviewed drafts of the article, and approved the final draft.
- Hassan Ziada conceived and designed the experiments, authored or reviewed drafts of the article, and approved the final draft.
- Eduardo Jorge Da Silva conceived and designed the experiments, authored or reviewed drafts of the article, and approved the final draft.
- Amit Jotangia conceived and designed the experiments, authored or reviewed drafts of the article, and approved the final draft.
- Sebastian Gabrielsson conceived and designed the experiments, authored or reviewed drafts of the article, and approved the final draft.
- Silje Gustafsson conceived and designed the experiments, authored or reviewed drafts of the article, and approved the final draft.
- Lynn Priddis conceived and designed the experiments, authored or reviewed drafts of the article, and approved the final draft.

## Human Ethics

The following information was supplied relating to ethical approvals (*i.e.*, approving body and any reference numbers):

Edith Cowan University granted ethical approval for this study (Ethical application Ref: 2019-00741-ROGERS).

## Data Availability

The data is available at Figshare: Rogers, Shane (2023). Further development of the reflective practice questionnaire (data). figshare. Dataset. https://doi.org/10.6084/m9.figshare.22776251.v1.

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
