# Peer review of "Further development of the reflective practice questionnaire"

_PeerJ, doi:10.7717/peerj.16879_

## Round 0.1 · original submission · Minor Revisions

Thank you for submitting to PeerJ. Please address multiple issues raised by the reviewers, especially regarding the introduction and discussion. Meanwhile, please provide a scatterplot of the correlation between RPQ reflective practice measure with the SRIS self-reflection and RRQ self-reflection in order to show that the associations are genuine.

Reviewer 1 ·

Basic reporting

The article is a comfortable read and easily understandable across professions.

Experimental design

No issues

Validity of the findings

Validity of the findings secure the premise of a decreased volume of questions. Perfect idea and strengthen the RPQ to be even more generalizable across man profession.

Additional comments

This manuscript takes an interesting but important view of the RPQ that further solidifies its application in interprofessional application. The premise to shorten the questions makes the RPQ more readily accessible and useable for many different professions. Having a shortened, more flexible tool can make the RPQ more applicable across many professions.

·

Basic reporting

No comments

Experimental design

No comments

Validity of the findings

No comments

Additional comments

Reason why the response in likert changed from 1 = not at all, 2 = slightly, 3 = somewhat, 4 = moderately, 5 = very much, and 6 = extremely
to
Almost always scale (scoring: 1. Very rarely 2. Rarely 3. Sometimes 4. Often 5. Very often 6. Almost always)

·

Basic reporting

Minor changes required - see table of feedback

Experimental design

Minor clarification required - see table of feedback

Validity of the findings

Please see table below

Additional comments

Please see annottated PDF file attached with copmments for revision/consideration.

This is a great development in the field and will be particularly useful with clinical samples to assess ongoing effectiveness of reflective sessions, as a brief measure of reflective capacity. I will certainly consider how I can build this into ongoing evaluation of reflective sessions. The introduction and rationale is sound. It is also great to have been able to derive a single score measure. The comparison of the RPQ in this article, with other measures of reflection is also very helpful. Thank you for publishing this body of work!

Reviewer 4 ·

Basic reporting

The literature review is mostly clear and gives specific and copious evidence as to how and why the RPQ has been used (and shown to be useful) from past work. Line 110 that describes outcome of the Van Winkle (2022) but would be helpful to clarify what is meant by "broader activities". This is a minor point, but one that's of interest to educators and researchers. References are numerous and from varied sources. The links to the data , charts, and brief study worked and are appreciated.

Experimental design

Research is within the Aims and Scope of the journal and the research questions are relevant, given the authors report finding many researchers desired an single RPQ score to report on general self-reflection. I think it shows great diligence that the authors responded in professional formats to how others could use the existing earlier version of the tool to derive this kind of score, but then follow up again here with this study to endorse this revised format and idea further. This fills a clear gap and request that existed for those studying and teaching about self-reflection.

The Methods section is clear, however I think it would be helpful to address that (line 246) the desire for improvement sub-scale "has not appeared to have been of much interest/use" earlier in the introduction/literature review. It seems kind of thrown in here where new information is not usually introduced and makes it seem like a less rigorously vetted construct that was chosen for elimination.

Validity of the findings

The data respond very clearly to aims of the study. They are statistically sound in terms of the models and factor analyses. Conclusions are also logical, although I agree that researching this with students (versus those engaging in work as a primary daily occupation would have likely been more meaningful) was a limitation, but this was clearly stated. I appreciated the between-fields comparisons so researchers/faculty from these various areas have an idea of how accurately the tool will capture constructs for their populations. The authors suggest other areas for study, including the obvious need for tools that are not self-perception scales.

Additional comments

The Discussion section is a predominantly a summary of the results section. Tying results to previous work that has done this kind of tool revision would lend strength to the author's points and their methodology.

---

## Round 0.2 · accepted · Accept

Thank you for addressing the reviewers' concerns.

Reviewer 4 ·

Basic reporting

I think the article is much easier to read and understand now. Thank you for making the suggested edits!

Experimental design

No issues

Validity of the findings

No concerns

Additional comments

I appreciate the efforts the authors made to revise and create a very clear and useful study!